# Quantum Hamiltonian Descent for Rigid Image Registration

## Abstract

Energy-based formulations of problems in computer vision and in particular the image registration problem are notoriously non-convex and traditionally require a combination of various techniques to obtain a good solution of the underlying optimization problem during the inference step. In this work, we explore how a recent result from quantum computing can help with this task. Specifically, we show how to apply Quantum Hamiltonian Descent to an optimization problem occuring in image registration. Numerical simulations on real-world data show that the method allows to recover good global minima despite the strong non-convexity, without relying on any heuristics or meta-strategies for aiding the optimization.

## 1 Introduction

Many tasks in computer vision suffer from difficult to solve optimization problems. One such example is image registration (Modersitzki, 2009; Oliveira & Tavares, 2014). Given two images, the task is to find a transformation that aligns them with each other. We represent the *reference* (fixed) and *template* (moving) images $R$ and $T$ as functions $R, T \colon \Omega \to \mathbb{R}$ with $\Omega \subseteq \mathbb{R}^d$. The unknown transformation is modelled by a mapping $\varphi \colon \mathbb{R}^d \to \mathbb{R}^d$ that transforms point coordinates in the reference image to the template image coordinate system, ideally such that $T \circ \varphi \approx R$.

The problem as such is highly underdetermined and thus requires regularization. In energy-based registration, the transformations are typically found by solving optimization problems of the general form

$$\min_{\varphi \colon \mathbb{R}^d \to \mathbb{R}^d} \mathcal{D}(R, T \circ \varphi) + \mathcal{R}(\varphi), \tag{1}$$

where $\mathcal{D}$ is a distance measure between two images and $\mathcal{R}$ is a regularizer enforcing that $\varphi$ is "well-behaved" or "sensible".

Even including regularization, the resulting optimization problems are typically highly non-convex (Figure 1). Non-convexity is generally a non-desirable property for energy-based methods, which ultimately rely on iterative optimization methods that use local information—e.g., function values and derivatives—around the current iterate and are therefore prone to getting stuck in local minimizers. Therefore, these methods are often limited to local convergence results and require heuristics such as coarse-to-fine strategies or multi-step registration, for example by introducing an affine or feature point-based pre-registration.

One could hope that this situation would be improved by learning-based data-driven methods, and considerable effort has been made in this direction (Haskins et al., 2020). However, it has been greatly hindered by the fact that true ground truth is almost always unavailable for real-world image registration problems, and while some quantitative advances have been made, they are relatively small compared to other fields such as image segmentation. Moreover, classical energy-based methods seem to be less limited to a narrow training data distribution (Jena et al., 2024).

Quantum computing (Nielsen & Chuang, 2000) offers a potential path out of this dilemma: By working on wave functions, it effectively allows to evaluate functions at *superpositions* of multiple—and potentially *all*—possible input values, paying the cost of only one function evaluation. With this superposition, we have global information at hand to construct iterates. In fact, for quantum mechanics, it can be shown that particles (e.g. electrons or photons) always consider *all possible paths* the particle *could* take and are completely determined by them (Feynman et al., 1966). As

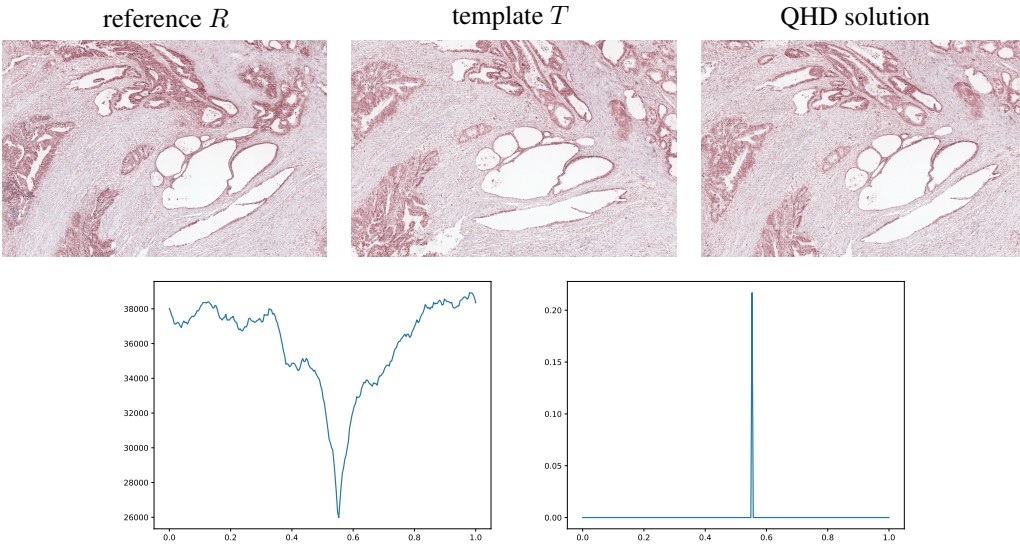

Figure 1: Rigid registration of a template image (top center) with respect to a reference image (top left) using QHD. While the energy landscape has a unique global minimum, its roughness requires considerable effort to prevent classical optimization methods from getting trapped in shallow local minimizers (bottom left, 1D slice). The quantum computing-based QHD converges to a distribution with a strong peak at the global minimizer (bottom right), so that subsequent sampling yields the optimal alignment (top right) with a high probability. Images from ANHIR dataset.

such, there is great potential for quantum mechanics in mathematical optimization. In this vein, Leng et al. proposed an optimization method called *Quantum Hamiltonian Descent* (QHD) (Leng et al., 2023) as a quantum-mechanical generalization of Gradient Descent. This method achieves global convergence even for non-convex functions and seems highly promising for application once powerful quantum computers become more available.

**Contribution.** In this work, we explore the potential of Quantum Hamiltonian Descent in image registration. Specifically,

- We review the required fundamentals of Quantum Mechanics and QHD;
- We discuss how to perform the integration of the required Schrödinger equation;
- We show how to apply the approach to image registration and provide numerical examples.

As is typical for current applications of quantum computing, our work should be seen as conceptual rather than pragmatic: The limitations in size and availability of current quantum hardware (Wilhelm et al., 2025) dictate that we restrict ourselves to numerical simulations. As the simulation cost increases exponentially with the size of the problem, we also limit ourselves to rigid registration. This is not a conceptual restriction, but it makes simulation feasible and also allows for visualization of the simulated quantum states. In particular, our transformations are parameterized by

$$\varphi(x; t, \alpha)^{-1} = R_\alpha x + t, \tag{2}$$

where $R_\alpha$ is the rotation matrix with angle $\alpha$ for which we assume the origin to be at the center of the image, and the vector $t \in \mathbb{R}^2$ is the translation component. Notice that we define $\varphi$ through its inverse here because if we want to visually achieve a rotation around the center, followed by a translation, we have to apply the inverse on the domain of the image. Besides making the simulation computationally tractable, the strong reduction in degrees of freedom also removes the need for an explicit regularizer $\mathcal{R}$.

Naturally, such simple models are far from the state of the art in terms of achieving the best image registration quality; however, the goal of this work is rather to explore new directions in solving the related optimization problems using quantum methods. Therefore, focusing on simpler models—

apart from being beneficial from a technical standpoint—is also justified in order not to confound the conclusions about the quality of the optimization process.

## 2 RELATED WORK

**Quantum optimization.** Various directions in quantum computing have been explored for optimization purposes, aiming to overcome challenges such as those posed by non-convex functions. Abbas et al. (2024) provide an overview of quantum optimization methods.

Among these approaches is quantum amplitude amplification on gate machines, motivated by the success of algorithms like Grover's search (Grover, 1996) or Shor's factoring (Shor, 1999). Amplitude amplification relies on interference to enhance the probability of measuring a desired solution state, while suppressing undesired states. The blueprint consists in initializing the quantum system in a superposition state of all possible solution, applying an oracle that evaluates the objective function on all the solution states, and a diffusion or amplification operator such as the Fourier transform to let the system interfere constructively in favor of the optimal solution.

Another direction is Adiabatic Quantum Computation (AQC), which leverages quantum annealing techniques for solving optimization problems (Kadowaki & Nishimori, 1998; Albash & Lidar, 2018). The idea is to initialize the system in the ground state of a simple Hamiltonian, for which the solution is easy to construct, and to then evolve the Hamiltonian toward a more complex structure that encodes the optimization problem. If the process follows the adiabatic theorem, the system remains in its ground state, ultimately leading to the optimal solution. Early applications of AQC have primarily been conducted using quantum annealers, focusing on combinatorial optimization problems.

Our work is primarily based on Leng et al. (2023), who rephrased the AQC framework for gate computers and continuous optimization problems, naming the method Quantum Hamiltonian Descent (QHD). This involves discretizing the time evolution of AQC an simulating it on a gate computer through a series of function oracle calls and applications of a well-designed diffusion operator. This diffusion operator is rooted in the quantum dynamics of particles and is tailored to amplify the amplitude of the solution state.

**Quantum computer vision.** Computer vision has recently started to explore quantum solutions. Early applications of quantum computing in computer vision predominantly focused on selection and permutation problems, which are inherently binary and combinatorial in nature. Key active research fields include model fitting (Farina et al., 2023; Pandey et al., 2025), graph and shape matching (Benkner et al., 2020; 2021; Bhatia et al., 2023), permutation synchronization (Birdal et al., 2021), k-means clustering (Zaech et al., 2024; Jaiswal, 2023), and object tracking (Zaech et al., 2022). These early works were designed for AQC and share a common approach: modeling the underlying problem as a Quadratic Unconstrained Binary Optimization (QUBO) problem, making them directly compatible with quantum annealing techniques.

Recently, a shift was observed to solutions relying on the gate-based paradigm. Yang et al Yang et al. (2024) proposed a Bernstein–Vazirani circuit implementation for robust fitting. K-mean clustering was tackled by Poggiali et al. (2024) on a gate computer. While there are only a few such gate-based prescriptive methods with well-designed circuits, which are challenging to construct, the trend is also shifting toward Quantum Machine Learning (QML) approaches that offer more generic and flexible solutions (Cerezo et al., 2021). QML-driven approaches leverage parameterized quantum circuits to generalize learning-based solutions across a wider range of applications. This includes, for example, image classification (Henderson et al., 2020; Jing et al., 2022; Kuros & Kryjak, 2022; Fan et al., 2023; Senokosov et al., 2024), implicit image representation (Zhao et al., 2024), generative models (Niu et al., 2022; Dallaire-Demers & Killoran, 2018), and point cloud auto-encoding and classification (Rathi et al., 2023; Baek et al., 2023).

Most related to ours are the works on registration and transformation estimation. Point-set registration methods (Golyanik & Theobalt, 2020; Meli et al., 2022; 2025) can be used for landmark-based image registration but require the additional step of collecting meaningful image landmarks beforehand. Braunstein et al. Braunstein et al. (2024) proposed to use quantum annealers as solvers for a Markov Random Field formulation of stereo matching. Chen et al. (2023) implemented rigid image

registration on a gate computer using a quantum Powell method to minimize the Sum of Squared Errors (SSE). However, this method still relies on a classical procedure for optimization, and the Powell method remains susceptible to the non-convexity of the problem, lacking guarantees for finding the global solution.

## 3 PRELIMINARIES

In quantum mechanics (Feynman et al., 1966, Section 2-4), the state is represented by a time-dependent normalized (norm-1) element $\psi_t$ in some complex Hilbert space $\mathcal{H}$. When $\mathcal{H}$ is a function space, these elements are also called *wave functions*. Their temporal evolution is governed by the *Schrödinger equation*

$$i\hbar \frac{\partial}{\partial t} \psi_t = H(t)\psi_t, \tag{3}$$

where $\hbar$ is the reduced Plank constant and $H(t)$ is a (densely defined) self-adjoint linear operator on $\mathcal{H}$, the *Hamiltonian*, representing the energy contained in the system. An example of a typical Hamiltonian is

$$H(t)\psi(x) = -\frac{\hbar^2}{2m} \Delta \psi(x) + V(x,t)\psi(x), \tag{4}$$

where the Hilbert space is chosen as $\mathcal{H} = L^2(\mathbb{R}^d)$, $\Delta$ is the Laplacian operator and $V(\cdot, t)$ is some map $\mathbb{R}^d \to \mathbb{R}$ representing the *potential energy* of the system at time $t$. Often, the units are chosen pragmatically such that $\hbar = m = 1$. The solution $\varphi_t$ at time $t$ can be written in terms of the linear solution operator $U$ according to $\psi_t = U(t, t_0)\psi_{t_0}$. As $H(t)$ is self-adjoint, $U(t, t_0)$ is necessarily unitary, so that it preserves the normalization of the states.

A peculiar property of quantum mechanics is that measuring a particle changes its state. Additionally, the result of the measurement is not deterministic but probabilistic. For the purposes of this work, we are only interested in measurements of the position of a particle. In that particular case, when $\psi$ is a normalized state in $L^2(\mathbb{R}^d)$, we have the equation

$$\int_{\mathbb{R}^d} |\psi(x)|^2 \mathrm{d}x = 1. \tag{5}$$

Thus, the squared absolute values of $\psi$ form a probability distribution over the space $\mathbb{R}^d$. Measuring the particle's position yields a random $x \in \mathbb{R}^d$ with probability given by the density $|\psi(\cdot)|^2$, collapsing the wave function to the Dirac delta $\delta_x$ at the observed point.

Quantum computing (Nielsen & Chuang, 2000) is usually formulated in the finite-dimensional setting where $\mathcal{H} = \mathbb{C}^{2^n}$, so that each possible value $k \in \{0, \ldots, 2^n - 1\}$ of an $n$-bit register can be associated with the corresponding $k$-th unit vector $e_k \in \mathbb{C}^{2^n} =: |k\rangle$, using Dirac notation. Such a system is called a *quantum register* and thought to be comprised of $n$ *qubits*. Generically, its state can be written as a superposition of all possible values where the squared amplitudes $\alpha_k$ sum to one:

$$|\psi\rangle = \sum_{k=0}^{2^n-1} \alpha_k |k\rangle, \quad \sum_{k=0}^{2^n-1} |\alpha_k|^2 = 1. \tag{6}$$

When constructing quantum computing algorithms for solving minimization problems such as (1), one first discretizes the search space for $\varphi$ using $n$ qubits. The art then lies in finding ways to enhance the amplitudes of the desired outcomes $k$—i.e., the ones corresponding to minimizers— relative to other amplitudes, so that a subsequent measurement yields a global minimizer of (1) with high probability.

## 4 QUANTUM HAMILTONIAN DESCENT

Quantum Hamiltonian Descent (QHD) as introduced in Leng et al. (2023) is a quantum algorithm for solving generic optimization problems

$$\min_{x \in \mathbb{R}^d} f(x) \tag{7}$$

given $f : \mathbb{R}^d \to \mathbb{R}$. The authors build on the Bregman-Lagrangian framework (Wibisono et al., 2016), which formalizes the Gradient Descent method as a problem in classical Lagrangian mechanics, and directly translate it into quantum mechanics. As a result, they propose a quantum system determined by the Hamiltonian

$$H(t) = e^{\varphi_t}\left(-\frac{1}{2}\Delta\right) + e^{\chi_t} f \tag{8}$$

with suitable real parameters $\varphi_t, \chi_t$. Under the condition that $e^{\varphi_t}/e^{\chi_t} \overset{t\to\infty}{\to} 0$ and certain regularity assumptions on $f$ and the Hamiltonian, they achieve the convergence result (Leng et al., 2023, Chapt. B.2, Thm. 2)

$$\lim_{t\to\infty} \mathbb{E}[f]_{\psi_t} = \min f, \tag{9}$$

where $\mathbb{E}[f]_{\psi_t}$ is to be understood as the expectation value for $f$ evaluated at the measurement of the position of the particle in state $\psi_t$. Consequently, if one can construct a system with the Hamiltonian (8), measuring its state after a sufficiently long time $t$ will yield a near-optimal solution $x$ with high probability.

**Integrating the Schrödinger Equation.**    In this section, we derive a numerical solution to the Schrödinger equation (3) following Leng et al. (2023), applied to the case of rigid image registration.

In order to solve the rigid registration problem, the domain of the wave function should be the space of all possible parameters $(t, \alpha) \in \mathbb{R}^d, d = 3$, in (1). We discretize the domain on a regular grid with periodic boundary conditions, assigning $n$ qubits (i.e., $n$ bits of precision) to each of the $d$ registration parameters. As a result, the quantum state $\psi_t$ at any time $t$ resides in a Hilbert space of dimension $N := 2^{n\cdot d}$.

This discretization yields the Hamiltonian

$$\hat{H}(t) = e^{\varphi_t}\hat{L} + e^{\chi_t}\hat{F}, \tag{10}$$

where $\hat{L}$ is a Laplacian matrix (with absorbed factor $-\frac{1}{2}$) and $\hat{F}$ is a diagonal matrix containing all values of the function $f$ at each point of the grid.

Next, we temporally discretize the Hamiltonian $\hat{H}(t)$ with some time resolution $\mathrm{dt} > 0$. With this, we can explicitly solve the linear differential equation and the result is given by

$$\psi_{j+1} = \exp(-ia_j\mathrm{dt}\hat{L} - ib_j\mathrm{dt}\hat{F})\,\psi_j, \tag{11}$$

where $a_j = e^{\varphi_j\cdot\mathrm{dt}}$ and $b_j = e^{\chi_j\cdot\mathrm{dt}}$. To further simplify the computation, we employ the approximation

$$\exp(-ia_j\mathrm{dt}\hat{L} - ib_j\mathrm{dt}\hat{F}) \approx \exp(-ia_j\mathrm{dt}\hat{L})\exp(-ib_j\mathrm{dt}\hat{F}), \tag{12}$$

which is justified by the Trotter-Suzuki approximation (Nielsen & Chuang, 2000, Equation 4.103)

$$e^{\tau(A+B)} = e^{\tau A}e^{\tau B} + \mathcal{O}(\tau^2) \tag{13}$$

for $\tau \to 0$ when the time resolution $\mathrm{dt}$ is fine enough. As $\hat{L}$ is diagonalizable by the Fourier transform $\mathcal{F}$, that is, $\hat{L} = \mathcal{F}\hat{D}\mathcal{F}^{-1}$ with some diagonal matrix $\hat{D}$, Equation (12) allows us to further simplify

$$\psi_{j+1} \approx \mathcal{F}\exp(-ia_j\mathrm{dt}\hat{D})\mathcal{F}^{-1}\exp(-ib_j\mathrm{dt}\hat{F})\psi_j. \tag{14}$$

Thus, the final spatially and temporally discretized evolution to implement is

$$|\psi_{j+1}\rangle := \mathcal{F}\exp(-ia_j\mathrm{dt}\hat{D})\mathcal{F}^{-1}\exp(-ib_j\mathrm{dt}\hat{F})|\psi_j\rangle, \quad |\psi_0\rangle = \frac{1}{\sqrt{N}}\sum_{k=0}^{N-1}|k\rangle, \tag{15}$$

now using Dirac notation and choosing the initial state as the uniform superposition.

On a classical computer, following this process is of no use, as it would require to compute all values of $f$ on the grid to find $\hat{F}$, which has the same effort as brute-forcing the problem over all $N$ possible inputs. On a quantum computer, however, as long as a quantum implementation of $f$ is available, we can implement $e^{-ib_j\mathrm{dt}\hat{F}}$ using a single evaluation as the linear operator

$$|x\rangle \longmapsto e^{i\theta f(x)}|x\rangle \tag{16}$$

where we set $\theta = -b_j \mathrm{dt}$ in each iteration. The operation (16), the so-called *phase oracle* for functions $f \colon \{0, \ldots, N-1\} \to \{0, \ldots, N-1\}$ is a unitary transformation and can always be implemented on a quantum computer, though the explicit form of the corresponding unitary operator may be challenging to derive. Similarly, we implement the diagonal $e^{-ia_j \mathrm{dt} \hat{D}}$.

The remaining Fourier transform $\mathcal{F}$ and its inverse can be efficiently implemented using the Quantum Fourier Transform (QFT) (Nielsen & Chuang, 2000, Chapter 5), which, on a quantum computer, has a complexity of $\mathcal{O}(m^2)$ on $m$ qubits. This is even faster than the Fast Fourier Transform on an equivalent $2^m$-dimensional vector in classical computation, which has a complexity of $\mathcal{O}(m 2^m)$. We apply the QFT independently to each of the $d$ sub-registers of the state vector $\psi$, yielding a total complexity of $\mathcal{O}(n^2)$.

## 5 A NOTE ON THE PHASE ORACLE IMPLEMENTATION

The crucial problem-specific step in the QHD approach is being able to efficiently evaluate the phase oracle (16), i.e., to provide a quantum implementation of the objective. Fortunately, in theory any classical function oracle defined on binary inputs can be implemented efficiently as a quantum circuit, which was one of the original motivations for the quantum circuit model (Nielsen & Chuang, 2000, Chapter 1.4.1). Moreover, it is a standard result that any such quantum oracle can be transformed into a phase oracle (Childs, 2004, Rule 1.6). Consequently, our method can be implemented on any circuit-based quantum computer.

However, it is prudent to confirm that this is actually practical without having to resort to classical computation. Here, we demonstrate a fully working QHD circuit for registering the $2 \times 2$ images shown in Figure 2, focusing on the 2D translation parameters for simplicity. As it only requires 13 qubits, it can be simulated on current hardware, allowing to verify the approach end-to-end.

The circuit comprises five registers: a register $|\cdot\rangle_R$ for the reference image $R$, a register $|\cdot\rangle_T$ for the template image $T$, a register $|\cdot\rangle_{\mathrm{SSD}}$ for the Sum of Squared Differences (SSD), and two $m$-qubits registers $|\cdot\rangle_{t_x} |\cdot\rangle_{t_y}$ encoding the translation parameters $t_x$ and $t_y$. We ensure that the SSD register has sufficiently many qubits to hold the SSD value in binary notation. For QHD, only the translation registers $|\cdot\rangle_{t_x}$ and $|\cdot\rangle_{t_y}$ matter as variables of the objective function; the other three registers act as ancillas, which must be reset at each iteration.

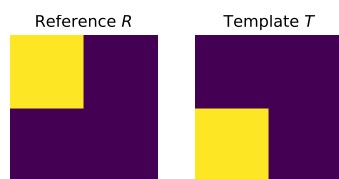

Reference $R$      Template $T$

Figure 2: Test images for QHD demonstration for translation.

We begin by preparing the translation registers in a uniform superposition, following Equation (15):

$$|\psi_0\rangle = \sum_{t_x, t_y} |0\rangle_R |0\rangle_T |0\rangle_{\mathrm{SSD}} |t_x\rangle_{t_x} |t_y\rangle_{t_y}. \tag{17}$$

Each QHD iteration proceeds as follows: (i) Encode the reference and template images as plain binary strings in their respective registers. (ii) Apply rigid deformations controlled by the translation qubits; those are controlled cyclic shifts on the row and column qubits of the template image. (iii) Compute the SSD by iterating over all pairs $|a\rangle |b\rangle$ of pixels in $R$ and $T$ at corresponding positions and updating the SSD register as $|a\rangle |b\rangle |\mathrm{SSD}\rangle \mapsto |a\rangle |b\rangle |\mathrm{SSD} + (a^2 + b^2 - 2ab)\rangle$, using quantum arithmetic. This yields the state

$$|\psi_0'\rangle = \sum_{t_x, t_y} |R\rangle_R |T \circ \varphi(t_x, t_y)\rangle_T |\mathrm{SSD}(t_x, t_y)\rangle_{\mathrm{SSD}} |t_x\rangle_{t_x} |t_y\rangle_{t_y}. \tag{18}$$

Next, we encode the SSD values as phases. Writing one such value as $|\mathrm{SSD}\rangle_{\mathrm{SSD}} = |q_0 q_1 \cdots q_K\rangle_{\mathrm{SSD}}$, the phase gate sequence $\bigotimes_{i=0}^{K} P\left(-b_0 \cdot \mathrm{dt} \cdot 2^i\right) |q_0 q_1 \cdots q_K\rangle_{\mathrm{SSD}}$ imprints the SSD value in the phase of the state. By phase kickback (Cleve et al., 1998), this applies the phase directly to the translation registers $|t_x\rangle_{t_x} |t_y\rangle_{t_y}$. Finally, we un-compute (Aaronson, 2003; Aaronson et al., 2015) the SSD evaluation, the rigid deformation, and the image encoding, thereby restoring the ancilla registers to $|0\rangle$ and leaving them ready for the next iteration:

$$|\psi_0''\rangle = \sum_{t_x, t_y} e^{-ib_0 \cdot \mathrm{dt} \cdot \mathrm{SSD}(t_x, t_y)} |0\rangle_R |0\rangle_T |0\rangle_{\mathrm{SSD}} |t_x\rangle_{t_x} |t_y\rangle_{t_y}. \tag{19}$$

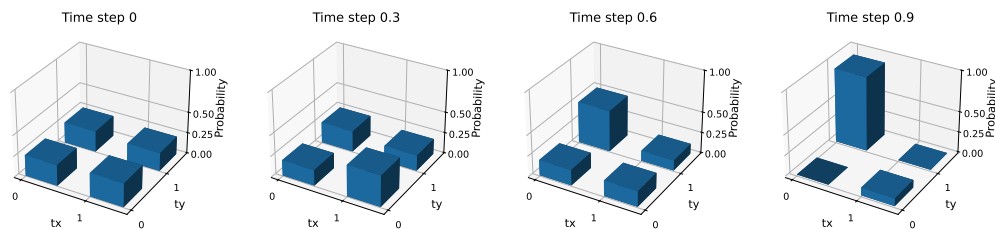

Figure 3: Snapshots of the state vector restricted to the translation registers $(t_x, t_y)$ at selected time steps of QHD for the input images in Figure 2. The correct translation parameter accumulates probability mass as iterations progress, illustrating the amplification effect of the algorithm.

Subsequently, we exponentiate the Laplacian by applying the inverse QFT, a diagonal unitary formed from the precomputed Laplacian eigenvalues (scaled by $a_0 \cdot \mathrm{dt}$), and the forward QFT. This yields

$$|\psi_1\rangle = \frac{1}{4} \sum_{u_x, u_y} \left[ \sum_{t_x, t_y, k_x, k_y} e^{-ib_0 \cdot \mathrm{dt} \cdot \mathrm{SSD}(t_x, t_y) - ia_0 \cdot \mathrm{dt} \cdot \lambda(k_x, k_y) + 2\pi i \frac{k_x(u_x - t_x)}{2^m} + 2\pi i \frac{k_y(u_y - t_y)}{2^m}} \right]$$
$$|0\rangle_R |0\rangle_T |0\rangle_{\mathrm{SSD}} |u_x\rangle_{t_x} |u_y\rangle_{t_y}, \qquad (20)$$

which serves as the input for the next iteration, with $u_x, u_y, k_x, k_y$ arising from the QFT and inverse.

We simulated the circuit, tracking the state vector evolution (Figure 3). In practice, we found the algorithm more stable when scaling the SSD values by a factor of $c$ in the phase gates, i.e., $P(b_0 \cdot s \cdot c \cdot 2^i)$ which balances the SSD function values against the Laplacian eigenvalues. For the experiments we used $c = 4$. The remaining parameters were chosen as $a_j = 2/(s + t^3)$, $b_j = 2 \cdot t^3$, $\mathrm{dt} = 0.1$, and $T = 1$, yielding 10 QHD iterations on 13 qubits. Figure 3 validates the approach: QHD successfully amplifies the measurement probability of the correct translation parameters.

While small, the scale of this example is close to the limit of current hardware. Doubling either the image resolution or the grayscale resolution already raises the qubit count beyond 32, which exceeds our current simulation capabilities. This motivates the exploration of more advanced image encodings and optimized arithmetic operations—directions we leave for future work. Importantly, these challenges should not be viewed as obstacles to the algorithm itself.

## 6 RESULTS

For our experiments, we extracted regions from parts of a dynamic scenes dataset (Yoon et al., 2020) and histological images from the ANHIR dataset on Grand Challenges (Borovec et al.; Fernandez-Gonzalez et al., 2002; Gupta et al., 2018; Mikhailov et al., 2018; Bueno & Deniz, 2019). For the dynamic scene images, we selected images from the same frame but from different camera perspectives. The histological image data consists of two different slices of the same tissue with different staining. Note that in particular on the dynamic scene images, perfect alignment using rigid deformation is impossible and not the goal, as we benchmark the performance of QHD and not the quality of the registration model used. However, such models are still useful and often used as preprocessing for sophisticated algorithms.

The numerical simulations of (15) were implemented in PyTorch 2.5.1 and performed on a 24-core AMD EPYC 74F3 with three NVIDIA A100 accelerators and CUDA 12.6. The diagonal $\hat{F}$ is fully evaluated and cached before running the actual algorithm.

We discretized the parameter space using eight bits for each of the three unknowns, resulting in a search space of size $2^{24}$ and a total qubit count of $n = 24$. We simulated the Schrödinger equation up to $T = 1$ time units with $r = 3000$ time steps, resulting in a time resolution of $\mathrm{dt} = 1/3000$. For the distance $\mathcal{D}$ in (1) we used the sum of squared differences (SSD). Notably, $T = 1$ is considerably shorter than in Leng et al. (2023). We believe that the fact that this shorter time suffices is due to a scaling effect, as the SSD admits much larger values than the examples in Leng et al. (2023). Iterating longer than $T = 1$, while yielding slight improvements for some of the test cases, can cause

| Dataset | $p(x^*)$ | $p(\bar{x})$ | $\|x^* - \bar{x}\|_{\ell^1}$ |
|---|---|---|---|
| balloon3 | 52.55% | 56.68% | 3 |
| playground | 4.18% | 7.60% | 4 |
| skating | 25.38% | 41.47% | 5 |
| coard17a | 64.27% | 66.56% | 1 |
| coard17b | 79.62% | 79.62% | 0 |
| breast5a | 58.93% | 58.93% | 0 |
| breast5b | 41.61% | 41.79% | 1 |
| lung-lesion2 | 78.57% | 78.57% | 0 |

Table 1: Results of QHD on real-world data for various pairs of reference images $R$ and template images $T$. The numbers $p(x^*)$ and $p(\bar{x})$ refer to the total probabilities of obtaining a measurement within a 5-neighborhood of the global minimizer $x^*$ and of the (most likely) QHD solution $\bar{x}$, respectively. By $\|\cdot\|_{\ell^1}$, we denote the $\ell^1$ distance in the discretized $256 \times 256 \times 256$ parameter grid.

reference $R$          template $T$          QHD solution          global optimium

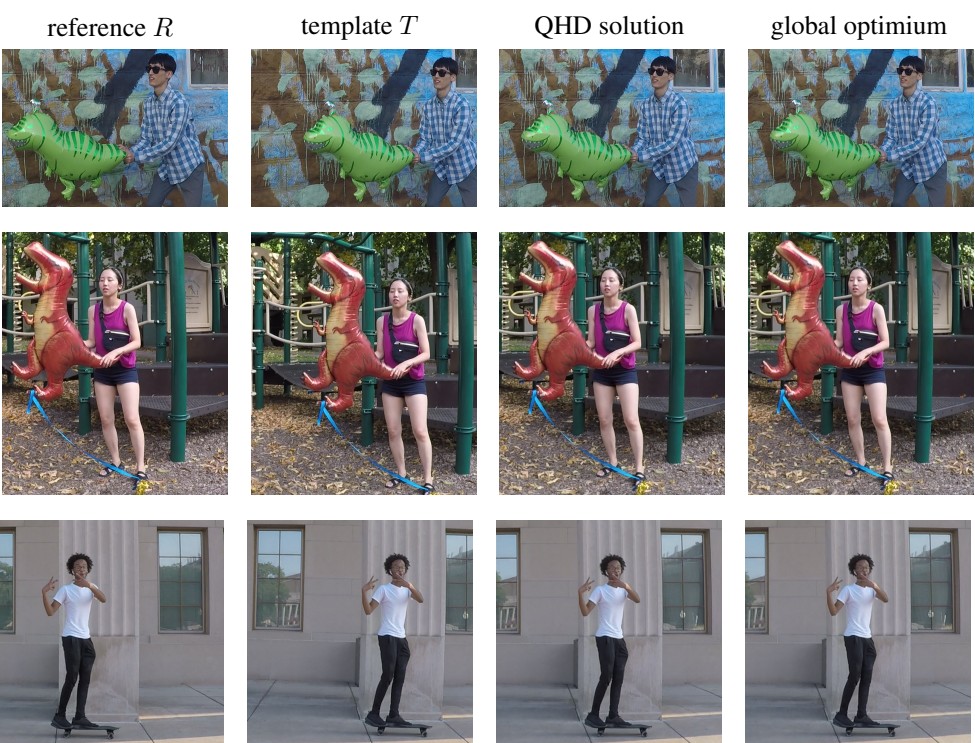

Figure 4: Test images and QHD registration results, in the same order as in Table 1. In all these cases, QHD finds a solution close to the exact global solution of the optimization problem. Note that the fact that in some cases (e.g., top row) the solution does not visually match the reference perfectly is a limitation of the model, which is not the focus of this work. Importantly, QHD finds excellent solutions to the optimization problem (second to the right and right column).

instability in others. This is due to discretization artifacts and can be mitigated by increasing the space and time resolution. Additionally, we originally tried to interpret the domain of the parameter space as the interval $[0, 1]^3$ as in Leng et al. (2023). However, for our energies, the resulting grid spacing of $h = 1/(2^8 - 1)$ was too unstable, which is why we pragmatically chose its square root. In general, optimally choosing the QHD discretization constants for a given objective function is still an open topic of research, see e.g. Leng & Shi (2025, Section 5.1).

In order to obtain a quantitative indication of the performance, we computed the probability of obtaining a measurement within an $\ell^1$ distance of at most 5 points from the global minimizer in the discretized $256 \times 256 \times 256$ parameter grid. Additionally, we determined the point $\bar{x}$ whose 5-neighborhood has the highest overall probability and computed its distance from the minimizer. We consider this as the solution of the QHD method. The numerical results are shown in Table 1 and the visual results in Figures 4 and 5.

reference $R$      template $T$      QHD solution

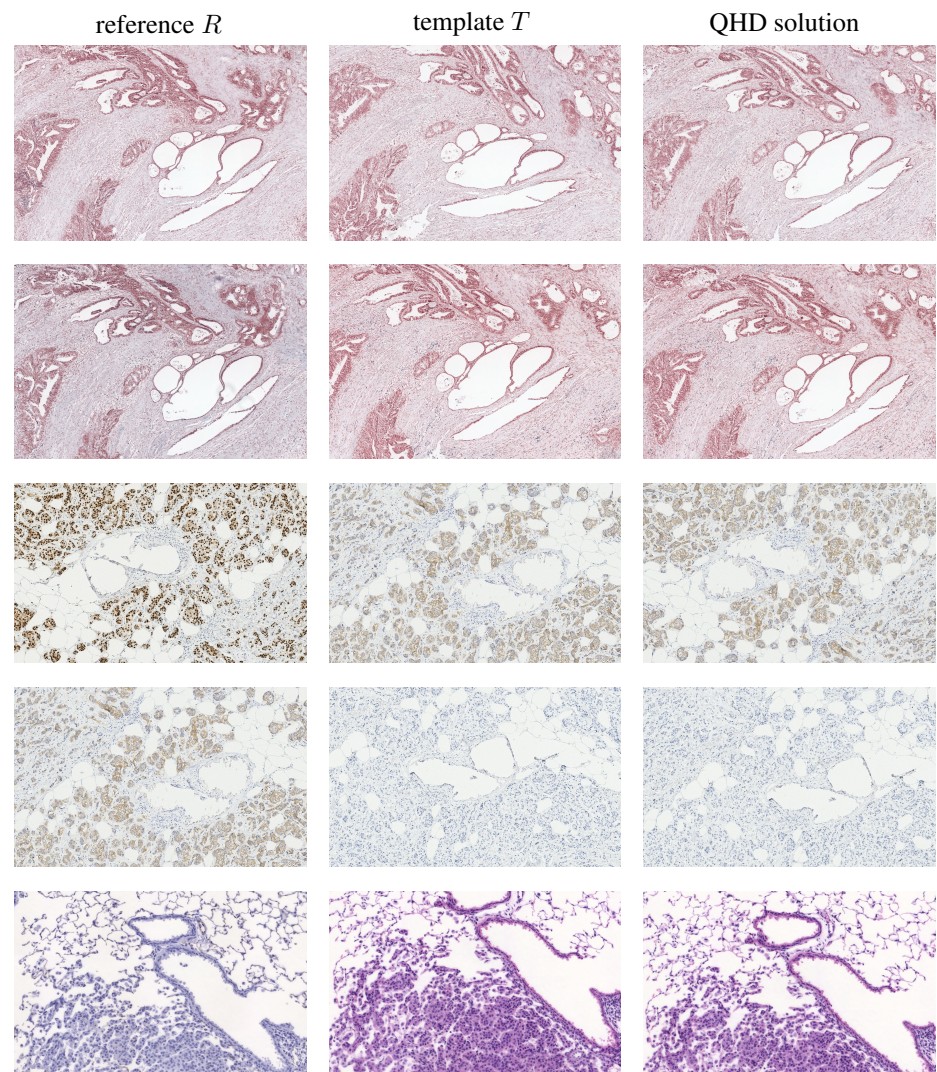

Figure 5: Test images and QHD registration results, in the same order as in Table 1. In all these cases, QHD finds a good solution that aligns well with the reference image despite the strong texture and corresponding roughness of the energy.

In all of our benchmarks, the QHD solution is close to the true global minimizer and is contained in its 5-neighborhood. Encouragingly, the probability of measuring the globally optimal solution is generally in the double digits and for many the test cases well above 50%. This should particularly be seen in the context of a search space of size $2^{24}$, where a uniform distribution would yield a probability in the order of $10^{-7}$.

**Conclusion.** While clearly the state of quantum hardware and simulation capabilities does not currently permit to practically outperform classical methods for image registration, we were excited to see that—at least in simulation—one of the harder problems of image processing can be successfully tackled with a relatively straightforward quantum-based approach. It will be interesting to see if in the future such methods will be able to obviate more preprocessing steps and processing pipelines.

**Reproducibility Statement.** The code for the implementation is available at [redacted for review]. It also contains the code used to generate the exact plots and benchmarks in this work. Once the datasets are downloaded from the original sources and placed at the appropriate locations, the code is fully functional. Instructions are provided in the README of the repository.

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
