# OpenReview forum: "Quantum Hamiltonian Descent for Rigid Image Registration"
_ICLR.cc/2026/Conference — ICLR 2026 Conference Withdrawn Submission_

### Official Review · Reviewer_Xb9t · 2025-10-30

**Soundness:** 3
**Presentation:** 3
**Contribution:** 3
**Rating:** 2
**Confidence:** 4

**Summary:**

The paper suggests the use of QHD to solve the highly non-convex problem of rigid image registration.
It shows that the solution can be obtained even for highly non-smooth and non convex problems.

The use of QHD is novel for this application and the numerical experiments are somewhat convincing

**Strengths:**

The paper suggests to solve a difficult problem using new tools in quantum computing. This can be one interesting application for quantum computers.

**Weaknesses:**

I am not sure about the fit to ICLR. Where is the learning?
This is an optimization paper and in my mind it should appear in an optimization journal, such as numerical optimization and applications.

Furthermore, the data sets are very easy. It seems that you are using FAIR (at least in notation). Look at the FAIR data. For example the matching of hands.
Very large deformations can be very important here. The deformations you have do not seem to be very large.
I would be interested in building a synthetic example, where the rotation/translation is known and see if you get the parameters.
Also it is not clear to me how the method perform when there is "noise". I do not mean Gaussian noise I mean elastic deformations that the model cannot capture

**Questions:**

see above

---

### Official Review · Reviewer_Vh6C · 2025-10-31

**Soundness:** 1
**Presentation:** 2
**Contribution:** 1
**Rating:** 0
**Confidence:** 4

**Summary:**

The work investigates image registration in the theoretical context of quantum computing. It applies the recently introduced QHD algorithm as a potentially robust solution for non-convex optimization problems in rigid alignment. The authors construct the QHD method for the image domain and show simulated optimization results on various paired 2D image datasets.

**Strengths:**

The underlying motivation is mostly sound: exploring early potential of quantum computing applications, where understanding realistic limits and use cases is important. But beyond the conceptual motivation, the paper offers limited insight.

**Weaknesses:**

Explicitly mentioned by the authors, this is a fully conceptual work. It offers minimal (to no) practical utility for image registration until, possibly, appropriate hardware becomes available in the future. While arguments of theoretical value *can* be useful, the work offers little to suggest QHD for rigid alignment actually has realistic future promise, even in a simulated context.

The authors' core argument is that by evaluating the loss function at all superpositions at once, the quantum implementation could converge better to global minima than current sensitive optimization approaches, but they provide no experiments to show this. This study requires baseline comparisons of registration accuracy, e.g. to classical gradient descent at the *absolute minimum*. The metrics in Table 1 have no standalone value without comparative statistics. Substantial experimental rigor is necessary to make the work at all convincing.

Image registration is a mature field, with many optimization baseline techniques (beyond simple gradient descent) that offer robust and fast solutions to best-navigate the non-convex loss landscape. Also, it's important to consider that in many cases, global minima do not always correspond to ground truth alignment parameters, especially in complex multimodal scenarios (e.g. that are very common in medical imaging). A major advantage of learning-based registration techniques is that they don't need to rely on possibly misleading evaluation functions at inference. QHD does not offer a solution to this problem.

The bulk of the paper text reviews quantum computing background and derives the previously established QHD method. It is very difficult to determine what technically novel contributions the authors are offering. This is ultimately an image registration application paper, yet there is little discussion of the current, well-established state of registration methods, realistic contextualization, or empirical evidence demonstrating the advantages of the proposed theoretical approach over existing current methods.

**Questions:**

If the proposed method runs on hardware that does not yet exist, but numerical simulations are currently too constrained to run reasonable comparisons to existing methods (line 477), then what is the scientific goal of this work?

How does it compare against classical gradient descent? How about multi-scale methods?

Does this work support realistic future directions and explorations that do not require the existence of useable quantum computers?

---

### Official Review · Reviewer_inN8 · 2025-11-02

**Soundness:** 2
**Presentation:** 2
**Contribution:** 1
**Rating:** 2
**Confidence:** 5

**Summary:**

The authors employ the quantum Hamilton descent to address the rigid image registration problem, a non-convex optimization challenge.

**Strengths:**

This is a pure experiemtnal manuscript.  I have not see any strengths.

**Weaknesses:**

I am not sure the experiments are right.

**Questions:**

Could you explain your experiment more?

---

### Official Review · Reviewer_EDQs · 2025-11-03

**Soundness:** 3
**Presentation:** 2
**Contribution:** 3
**Rating:** 4
**Confidence:** 2

**Summary:**

First let me remark that I know the basics of quantum computing but am not an expert in this field. My field is classical optimization and computer vision. Given this, my understanding of the paper is as follows.

The authors consider the problem of image registration, where we want to find a 2d-to-2d transformation that takes a given template image as close (in some metric) to a given reference image. The set of possible transformations is restricted, e.g., by giving this set explicitly or by a regularizer (i.e., a prior). This in general leads to a non-convex optimization problem. The authors aim to demonstrate that/how this problem can be solved by a method called Quantum Hamiltonian Descent (QHD), recently proposed by Lang et al 2023.

QHD is a quantum method for unconstrained continuous optimization, intended to be a quantum counterpart of classical gradient descent (in fact, a whole family of its accelerations, as proposed by Wibisono et al 2016). For a given objective function $f: \mathbb R^n\to\mathbb R$, QHD is instantiated by a time-varying Hamiltonian that is a weighted average of the laplacian (ie, kinetic energy) and potential field (the objective function). The weights change in time so that the kinetic energy gradually diminishes relative to the objective, so that the probability distribution given by the squared wave function gradually concentrates around global optimum.

To make this aim feasible, the authors consider only rigid transformations, parameterized by 3 real numbers (angle and translation). The challenge is now to instantiate QHD for this task. The Hamitonian is discretized in space, by encoding each of the 3 parameters $x$ (angle, translation) by 8 qubits. This turns the laplacian into a symmetric matrix and the potential into a diagonal matrix (with all the values $f(x)$ on its diagonal). By discretization in time, a recurrent formula is given, approximating the solution of the Schrodinger eqn.

This is simulated on a classical computer for experimental image pairs, taken form public datasets (outdoor scenes taken by a camera from two different places, medical image pairs). The results reported show that the method always ended up very near the global minimum. The experiments aim only to demonstrate that the method works - using only a few image pairs and not comparing with other registration methods.

Besides this, in section 5 the authors focus on actual possible implementation on a quantum computer. Here, the potential term of the hamiltonian (the objective function) is represented by a phase oracle, whose instantiation is challenging and problem specific. The authors show how thiis can be done (on a gate quantum computer) for images of size 2-by-2 pixels with periodic boundaries. This very small example needs 13 qubits (for any larger images we'd get beyond abilitis of current quantum HW). This curcuit is simulated on a classical computer, showing correct result for the 2-by-2 image pair.

**Strengths:**

It shows in detail how a relatively complex computer vision optimization problem can be solved by a quantum computer, though not yet nowadays for realistic image sizes.
The technical part and math seems correct to me.

**Weaknesses:**

I do not fully understand what exactly was done and what parts of the paper are novel.

Section 3 is clearly not novel, it is a recap of Schroedinger eqn.

Section 4 is mostly a general recap of a part of the QHD paper (Leng et al 2023), only a small part of the section is specific to the problem at hand.

Section 5 seems to contain most novelty: it proposes an actual implementation of QHD on a quantum computer (though this is not tested on a quantum computer but only simulated classically). However I am confused here: Leng et al 2023 propose to implement QHD on a D-Wave hardware, discretizing the Hamiltonian using a quantum Ising model. But here a universal gate quantum computer is assumed, to my understanding. The reason for this decision is not discussed in the paper.

To my understanding, to obtain the experimental results on realistic image pairs in Section 6, the (discretized) Schrodinger eqn is solved by simulation on a classical computer. Here, the matrix $\hat F$, which has on its diagonal all the objective values $f(x)$, is explicitly formed, right?

To summarize this part, it may be helpful if the authors were more explicit about contributions and novelty. Note, the first of the three contributions stated on lines 88+ is clearly not novel. The second contribution (how to integrate Schroedinger eqn) is also not novel, I believe, as this was already discussed in the QHD paper. So only the third item remains.

Next, I would appreciate some discussion about scalability of the method, on possible quantum hardware in near future. Is there a chance that in a near enough future it will be possible to run the method on quantum HW for realistic image size and wider classes of image transformations (non-rigid ones)? Perhaps, a table with the required numbers of qubits for various cases would be helpful. Currently, the approach may look too far from anything realistic.

Minor remark: if the images have high-frequency texture, the global minimum of the used objective function may not best correspond to ground truth transformation. Then it might be a good idea to smooth the images slightly before comparison, use a coarse-to-fine approach, etc. But this is not critical for this paper, which only aims to prove the concept.

**Questions:**

1) What is the motivation to assume circuit-base quantum computer, as opposed to the quantum Ising model assumed by Leng et al 2023?
More generally, compare the approach proposed in Leng et al with your approach.

2) What do you consider the most novel/valuable part of the paper?

---

### Official Review · Reviewer_e9Sk · 2025-11-06

**Soundness:** 3
**Presentation:** 4
**Contribution:** 2
**Rating:** 2
**Confidence:** 5

**Summary:**

This article proposes to use recently formulated Quantum Hamiltonian Descent to solve the rigid image registration problem. As formulated in the article, rigid registration is a non-convex problem that is costly (but not impossible) to solve on classical computers but can be formulated to work more efficiently on a subtype of quantum computers that leverage quantum annealing techniques, called Adiabatic Quantum Computers (AQCs). Crucially, these computers have been implemented in hardware, albeit at a small scale.

The authors propose a formulation that could be made to work on AQCs. They simulate a larger-scale AQC with significant but classical hardware, and show that their formulation can indeed solve the rigid image registration problem with good accuracy.

**Strengths:**

The paper is very clear and well illustrated. It presents its goal, challenges, contribution and limitations in an honest fashion. The physics behind AQCs and the way useful computations can be performed on them is enlightening.

The formulation is mathematically sound and the simulations are interesting.

**Weaknesses:**

The limitations on current AQC hardware are so severe that only trivial binary registration on tiny images could be envisaged today. There is no timeline on when more practical hardware could see the light of day. In its current form, it could be a long time indeed.

Given this, the simulation on classical hardware is an inefficient way to solve practical problems. As an illustration, the rigid image registration problem can be formulated in a convex way and is solved very efficiently on current classical (actually, the more practical and interesting affine registration problem is).

Given all this, the article mostly illustrates that even with non-existent prototyping AQCs, their practical use is of limited interest and could only be available in the non-foreseable future. If that was the conclusion of the article, I would be inclined to accept it, but in its current form, it is inconclusive.

**Questions:**

- Is there a timeline on AQC progress?
- When would a practical AQC be available capable of solving the image registration problem non trivially?
- The relationship to machine learning is weak. How could crucially important regularisation techniques be implemented on AQCs?

---

### Note · Authors · 2025-11-28

**Comment:**

Naturally we do not agree with most of the reviews, but have to accept that there appears to be not enough interest from the ICLR reviewer community in this pioneering technology, where important foundational research is happening at a level that obviously cannot yet compete with current classical methods. Therefore we withdraw the paper.

**Withdrawal Confirmation:**

I have read and agree with the venue's withdrawal policy on behalf of myself and my co-authors.